# Effects of Chronic Exposure to Low Levels of Dietary Aflatoxin B_1_ on Growth Performance, Apparent Total Tract Digestibility and Intestinal Health in Pigs

**DOI:** 10.3390/ani11020336

**Published:** 2021-01-29

**Authors:** Junning Pu, Qinghui Yuan, Hui Yan, Gang Tian, Daiwen Chen, Jun He, Ping Zheng, Jie Yu, Xiangbing Mao, Zhiqing Huang, Junqiu Luo, Yuheng Luo, Bing Yu

**Affiliations:** Institute of Animal Nutrition, Sichuan Agricultural University, Yaan 625014, China; junningpu@163.com (J.P.); qhyuan2020@163.com (Q.Y.); yan.hui@sicau.edu.cn (H.Y.); tgang2008@126.com (G.T.); dwchen@sicau.edu.cn (D.C.); hejun8067@163.com (J.H.); zpind05@163.com (P.Z.); yujie@sicau.edu.cn (J.Y.); acatmxb2003@163.com (X.M.); zqhuang@sicau.edu.cn (Z.H.); 13910@sicau.edu.cn (J.L.); luoluo212@126.com (Y.L.)

**Keywords:** aflatoxin B_1_, growth performance, apparent total tract digestibility, intestinal health, pigs

## Abstract

**Simple Summary:**

Aflatoxin B_1_ (AFB_1_) is one of the most toxic mycotoxins compounds produced by *Aspergillus*, a common fungi contaminant in food and animal feed. Although there are many studies on AFB_1_, most of them are focused on the acute toxic effects of high-dose AFB_1_ ingestion. The symptoms of acute AFB_1_ mycotoxicosis are rarely observed in actual animal production. However, long-term exposure to low levels of AFB_1_ is common in swine production and may contribute to chronic diseases. Therefore, this study investigated the effects of chronic exposure to low levels of dietary AFB_1_ on growth performance, apparent total tract digestibility and intestinal health in pigs. We found that chronic exposure to low levels of dietary AFB_1_ suppressed growth performance, reduced apparent total tract digestibility and damaged intestinal barrier integrity in pigs, which could be associated with the decreased intestinal antioxidant capacity and the increased pro-inflammatory cytokine production. These results could provide new insights for future studies on the prevention and treatment of AFB_1_ poisoning.

**Abstract:**

This study aimed to investigate the effects of chronic exposure to low levels of dietary aflatoxin B_1_ (AFB_1_) on growth performance, apparent total tract digestibility and intestinal health in pigs. In a 102-day experiment, fourteen barrows (Duroc×Landrace×Yorkshire, initial BW = 38.21 ± 0.45 kg) were randomly divided into control (CON, basal diet) and AFB_1_ groups (the basal diet supplemented with 280 μg/kg AFB_1_). Results revealed that the AFB_1_ exposure decreased the final BW, ADFI and ADG in pigs (*p* < 0.10). AFB_1_ exposure also decreased the apparent total tract digestibility of dry mater and gross energy at 50 to 75 kg and 105 to 135 kg stages, and decreased the apparent total tract digestibility of ether extract at 75 to 105 kg stage (*p* < 0.05). Meanwhile, AFB_1_ exposure increased serum diamine oxidase activity and reduced the mRNA abundance of sodium-glucose cotransporter 1, solute carrier family 7 member 1 and zonula occluden-1 in the jejunal mucosa (*p* < 0.05). Furthermore, AFB_1_ exposure decreased superoxide dismutase activity (*p* < 0.05) and increased 8-hydroxy-2′-deoxyguanosine content (*p* < 0.10) in jejunal mucosa. AFB_1_ exposure also increased tumor necrosis factor-α, interleukin-1β and transforming growth factor-β mRNA abundance in jejunal mucosa and upregulated *Escherichia coli* population in colon (*p* < 0.05). The data indicated that chronic exposure to low levels of dietary AFB_1_ suppressed growth performance, reduced the apparent total tract digestibility and damaged intestinal barrier integrity in pigs, which could be associated with the decreased intestinal antioxidant capacity and the increased pro-inflammatory cytokine production.

## 1. Introduction

The occurrence of mycotoxins in foodstuffs for humans and animals has been constituted as a threat to international public health [1]. Aflatoxins are secondary metabolites produced primarily by *Aspergillus* [2]. Among aflatoxins identified, aflatoxin B_1_ is the most toxic contaminant in foods and feedstuffs, and is classified as a Class I carcinogen by the International Agency for Research on Cancer [3]. Aflatoxin B1 needs to convert to AFB1-8,9-exo-epoxide (AFBO) to exert toxic effects [4]. Aflatoxin B_1_ has been characterized as hepatotoxic, teratogenic, carcinogenic, and immunosuppressive [5]. Aflatoxin B_1_ contaminated feeds can cause animal poisoning, whose manifestation includes growth retardation, liver and kidney damage, oxidative stress, immune inhibition, and increased susceptibility to diseases [6,7,8]. In addition, AFB_1_ remains in human food through animal-derived products (such as animal tissues, milk, and eggs), which may pose a threat to human health [9]. Therefore, AFB_1_ has raised concerns globally in animal production and human public health.

The gastrointestinal tract (GIT) is not only an important organ for nutrient digestion and absorption, it also plays a key role in defending against pathogen infection [10]. Aflatoxin B_1_ is rapidly absorbed into the blood from the GIT, followed by an extensive transformation into metabolites in the liver [11]. The GIT is the first organ by which AFB_1_ comes into the bodies of humans and animals; thus, this toxin should exert greater toxic impacts on the intestinal tract compared with other organs [7]. Nevertheless, the effects of AFB_1_ on the intestinal tract are often neglected and inconclusive. Therefore, it is important and necessary to study the effects of AFB_1_ on the intestinal health of pigs.

Although much research is available about AFB_1_ in pigs, most of these reports are focused on acute toxicity following the consumption of high doses of AFB_1_, which is characterized by body weight reduction, liver and kidney injuries, and immunosuppression [12,13,14,15]. Actually, dietary AFB_1_ levels could be very low due to taking good care of dietary ingredients. The Food and Agriculture Organization of the United Nations has reported that the maximum tolerance level of pigs to aflatoxins is 200 μg/kg [16]. The United States has limited the AFB_1_ concentration in pig diets to 300 μg/kg, while in China, the maximum AFB_1_ in pig diets is 20 μg/kg [8]. Furthermore, according to the global survey of mycotoxins in feedstuffs [17,18], the dose of aflatoxins is divided into three categories: realistic doses (representative of field conditions, <300 μg/kg); occasional doses (unfavorable weather conditions, 300~2000 μg/kg); and unrealistic doses (unlikely to occur in nature, >2000 μg/kg) [19]. Therefore, the symptoms of acute AFB_1_ mycotoxicosis are rarely observed in actual animal production. However, long-term exposure to low levels of AFB_1_ is common and may contribute to chronic diseases.

Therefore, in this study, we sought to determine the effects of chronic exposure to low levels of dietary AFB_1_ on growth performance, apparent total tract digestibility and intestinal health in pigs, thereby providing a scientific basis for guidance on the production of healthy pigs.

## 2. Materials and Methods

All procedures involved in the study were approved by the Animal Care and Use Committee of Sichuan Agricultural University (Approval number: CD-SYXK-2017-015).

### 2.1. Aflatoxin B_1_ Production and Diet Preparation

Aflatoxin was produced by *Aspergillus flavus* (ATCC28539; purchased from the China Center of Industrial Culture Collection) via fermentation on sterile, polished rice. The mold strain was cultured on sterile potato dextrose agar and incubated at 28 °C for 5–8 days to obtain a uniform fungus spore suspension. Following this, 10 mL of the fungus spore suspension containing 106 spores/mL was transplanted to 80 g sterile rice in Erlenmeyer flasks, and incubated at 28 °C. After 5 days, the rice was immersed in chloroform to kill the fungi, and then ground into fine powder. The concentration of AFB_1_ in rice powder samples was detected by ELISA kits (Suwei Microbiology Research Co., Ltd., Wuxi, China). Finally, the dried AFB_1_-contaminated rice powder (equal to 180 mg/kg of AFB1) was added to the basal diet to obtain the desired level of AFB_1_ diet (approximately 280 μg/kg of AFB_1_).

The contents of mycotoxins (AFB_1_, zearalenone (ZEA), deoxynivalenol (DON), T2 and ochratoxin (OTA)) in basal diet and AFB_1_ diet were analyzed by high-performance liquid chromatography (HPLC: Shimadzu LC-10 AT, Shimadzu, Tokyo, Japan) method [18,20]. The minimum detection concentrations are 0.10 μg/kg for AFB_1_, 1.00 μg/kg for ZEA, 10.00 μg/kg for DON, 25.00 μg/kg for T2, and 0.21 μg/kg for OTA, respectively. Concentrations of various mycotoxins in basal diet and AFB1 diet are presented in Table 1. Finally, the contents of AFB1 in basal diet and AFB1 diet were 0.40 µg/kg and 286.60 µg/kg, respectively. Only AFB_1_ exceeded the regulatory guidance concentration of Chinese National Standard (GB 13078-2001), while other mycotoxins did not exceed the regulatory limits of Chinese National Standard (GB 13078.2-2006, GB 13078.3-2007, and GB 21693–2008).

### 2.2. Experimental Design and Animal Management

Fourteen barrows (Duroc × Landrace × Yorkshire, initial BW = 38.21 ± 0.45 kg) were randomly divided into the control (CON, basal diet) and AFB_1_ groups (the basal diet supplemented with 280 μg/kg AFB_1_), with 7 pigs per group. All pigs were housed in individual metabolism cages (0.7 m × 1.5 m) and were given ad libitum access to water and feed. The experiment lasted 102 days and consisted of 4 stages: 38 to 50 kg, 50 to 75 kg, 75 to 100 kg and 100 to 135 kg. The basal diet (Table 2) was formulated to meet the National Research Council nutrient requirements (NRC) [21]. The pigs were weighed individually on day 1 and 103, and feed intake was recorded daily. These values were used to calculate average daily gain (ADG), average daily feed intake (ADFI) and the ratio of feed to gain (F/G).

### 2.3. Sample Collection

The apparent total tract digestibility was determined during the last 4 days of each stage (average body weight of pigs at each stage = 74.36 ± 0.93, 102.50 ± 1.58, 128.71 ± 2.36 kg, respectively). During the period of digestibility determination, fecal samples from pigs in each group were collected and weighted daily. After weighing, 10 mL of 10% H_2_SO_4_ solution was added to each 100 g of fecal sample, and subsequently stored in plastic bags at −20 °C. At the end of the 4 day period, all fecal samples from each pig were thawed at room temperature and mixed thoroughly, and then dried at 65 °C for 48 h, after which they were ground to pass through a 1 mm screen and stored at −20 °C for chemical analyses. All experimental diets were sampled and stored at −20 °C until chemical analysis for crude protein, dry mater, crude fat, and gross energy. Blood samples were collected from pigs via the anterior vein on day 103 following an overnight fast. After centrifugation (3500× *g* for 10 min at 4 °C), serum samples were harvested and stored at −20 °C until analysis. Subsequently, all pigs were euthanized by electric shock, and the jejunal tissue was immediately collected. Mucosal samples from the middle jejunum were scraped and rapidly stored at −80 °C until analysis. In addition, an approximately 3 g digesta sample from the colon was stored at −80 °C for microbial DNA analysis.

### 2.4. Chemical Analysis

The apparent total tract digestibility of crude protein, dry mater, ether extract and gross energy was determined by the method of acid insoluble ash (AIA) [22]. The crude protein (method 990.03), dry mater (method 930.15) and ether extract (method 945.16) were measured according to the methods described by AOAC (1995) [23]. Gross energy was determined using an automatic adiabatic oxygen bomb calorimeter (Parr Instrument Co., Moline, IL, USA). The apparent total tract digestibility was calculated using the following formula: apparent total tract digestibility (%) = {1 − [(A1 × F2)/(A2 × F1)]} × 100, in which A1 = the AIA content of the diet, A2 = the AIA content of feces, F1 = the nutrient content of the diet and F2 = the nutrient content of feces.

### 2.5. Diamine Oxidase Activity in Serum

The activity of diamine oxidase (DAO) in serum was measured by using Diamine Oxidase Assay kit (Nanjing Jiancheng Institute of Bioengineering, Nanjing, China) according to the manufacturer’s instructions. All determinations were done in triplicate, and absorbance was measured using a multi-mode microplate reader (SpectraMax M2, Molecular Devices, Sunnyvale, CA, USA).

### 2.6. Antioxidant Parameters in Jejunal Mucosa

The mucosal sample of jejunum was homogenized in ice-cold physiologic saline (w/v = 1:9). After centrifugation (3500× *g* for 10 min at 4 °C), the mucosal supernatant was collected to determine antioxidant parameters. The jejunal mucosal antioxidant parameters including total antioxidant capacity (T-AOC), superoxide dismutase (SOD) and malondialdehyde (MDA) were measured by the commercial kits (Nanjing Jiancheng Institute of Bioengineering, Nanjing, China) combined with a UV–VIS Spectrophotometer (UV1100, MAPADA, Shanghai, China) according to the manufacturer’s instructions. The total protein concentration of supernatants was determined by using a protein assay kit (Nanjing Jiancheng Institute of Bioengineering, Nanjing, China).

### 2.7. 8-OHdG and PCO Concentrations in Jejunal Mucosa

The concentrations of 8-hydroxy-2′-deoxyguanosine (8-OHdG) and protein carbonylation (PCO) in jejunal mucosa were determined using commercially available pig ELISA kits (Chenglin Institute of Bioengineering, Beijing, China) according to the manufacturer’s instructions.

### 2.8. Total RNA Isolation and Gene Expression Analysis

Approximately 40 mg of jejunal mucosa were used for total RNA extraction using TRizol Reagent (TaKaRa, Dalian, China). Reverse transcription was performed according to the instructions of the PrimeScriptTM RT reagent kit (TaKaRa, Dalian, China). Real-time PCR was conducted in a QuanStudio™ 6 Flex Real-Time PCR System (Applied Biosystems, Foster, CA, USA), using SYBR^®^ Premix Ex Taq™ II (TaKaRa, Dalian, China). The primer sequences were listed in Table 3 and purchased form TaKaRa (Dalian, China). The real-time PCR cycling conditions were as follows: 95 °C for 30 s, 40 cycles of 95 °C for 5 s, and 60 °C for 30 s. The relative mRNA levels of target genes were calculated using the 2^−ΔΔCt^ method with β-actin as the housekeeping gene [24].

### 2.9. Bacterial DNA Isolation and Microbial Real-Time Quantitative PCR

Bacterial DNA in colonic digesta was extracted by using the Stool DNA Kit (Omega Bio-Tek, Doraville, CA, USA). All primers and probes were listed in Table 4 and designed following the previous report [25]. Microbial real-time quantitative PCR was performed in a QuanStudio™ 6 Flex Real-Time PCR System (Applied Biosystems, Foster, CA, USA). Briefly, the total bacteria was detected using SYBR^®^ Premix Ex Taq™ II reagent (TaKaRa, Dalian, China), and the *Bacillus*, *Lactobacillus*, *E. coli* and *Bifidobacterium* were detected using PrimerScriptTM PCR kit (TaKaRa, Dalian, China) following the previous methods [26]. Furthermore, for the quantification of bacteria, specific standard curves were generated by constructing standard plasmids as presented by Chen et al. (2013) [26].

### 2.10. Statistical Analysis

Each pig was considered as an experimental unit. Bacterial copies were transformed (log10) before statistical analysis. All data were expressed as means ± standard errors (SE) and were analyzed by one-way analysis of variance (ANOVA) by SPSS 20.0 software (SPSS Inc., Chicago, IL, USA). Student’s *t*-test was used in order to compare the means. *p* < 0.05 was considered as significant, 0.05 ≤ *p* ≤ 0.10 was considered as a tendency.

## 3. Results

### 3.1. Growth Performance

Pigs fed the AFB_1_ diet trended to decrease their final BW, ADFI and ADG across the whole experiment compared with the CON group (Table 5, *p* < 0.10). However, no significant difference in F/G was observed between AFB_1_ group and CON group (*p* > 0.05).

### 3.2. Apparent Total Tract Digestibility

Compared with CON group, pigs fed the AFB_1_ diet significantly decreased the apparent total tract digestibility of dry mater and gross energy at the 50 to 75 kg and 105 to 135 kg stages, and decreased the apparent total tract digestibility of ether extract at 75 to 105 kg stage (Table 6, *p* < 0.05). However, no significant difference in the apparent total tract digestibility of crude protein was observed (*p* > 0.05).

### 3.3. Relative mRNA Expressions of Nutrient Transporters in Jejunal Mucosa

Pigs fed the AFB_1_ diet had significantly decreased mRNA expression of *SGLT1* and *SLC7A1* in jejunal mucosa compared with those fed the CON diet (Table 7, *p* < 0.05).

### 3.4. Serum DAO Activity and Relative mRNA Expressions of Barrier Junction Related Genes in Jejunal Mucosa

The serum DAO activity in the AFB_1_ group was greater than that in CON group (Table 8, *p* < 0.05). Meanwhile, pigs fed the AFB_1_ diet showed significantly decreased mRNA abundance of *ZO-1* in the jejunal mucosa compared with CON group (*p* < 0.05).

### 3.5. Antioxidant Capacity

Compared with the CON group, pigs fed AFB_1_ diet showed significantly decreased activity of SOD (Table 9, *p* < 0.05), and tended to show increased content of 8-OHdG in the jejunal mucosa (*p* < 0.10). However, no significant effects of dietary AFB_1_ on the activity of T-AOC and the content of MDA and PCO in jejunal mucosal were observed (*p* > 0.05).

### 3.6. Relative mRNA Expressions of Inflammatory Related Genes in Jejunal Mucosa

Compared with the CON group, pigs fed the AFB_1_ diet showed a significantly increased mRNA abundance of *TNF-α* and *IL-1β* (Table 10, *p* < 0.05), and tended to show an increase in the mRNA abundance of *TGF-β* in the jejunal mucosa (*p* = 0.05). However, no significant effect of dietary AFB_1_ on the mRNA abundance of *IL-8* in the jejunal mucosal was observed (*p* > 0.05).

### 3.7. Bacteria Populations

Pigs fed the AFB_1_ diet showed a significantly increased *Escherichia coli* population incolonic digesta compared with the CON group (Table 11, *p* < 0.05). However, no significant difference was observed on the populations of total bacteria, *Lactobacillus*, *Bacillus* and *Bifidobacterium* in colonic digesta between the AFB_1_ group and the CON group (*p* > 0.05).

## 4. Discussion

Aflatoxin B_1_ (AFB_1_) is one of the most common mycotoxins found in feedstuffs such as corn, barley, and wheat [27]. Ingestion of AFB_1_ by animals causes many health issues including decreased feed intake and body weight, liver damage, immune suppression and even death, which eventually leads to significant economic losses [28,29]. Pigs are easily exposed to AFB_1_ due to the composition of their feed [30]. Previous studies have shown that piglets fed a diet contaminated with 200 μg/kg of aflatoxins decreased growth rate and feed intake [31]. Marin et al. (2002) reported a decrease in the growth rate of piglets fed a diet contaminated with 280 μg/kg of AFB_1_ [8]. Similarly, the data of this study revealed that the ingestion of a diet containing 280 μg/kg AFB1 trended to decrease the final BW, ADFI and ADG of pigs. The adverse effect of AFB_1_ on growth performance partially results from undereating and from decreased nutrient digestibility [32,33]. In this study, pigs fed the AFB_1_ diet showed significantly decreased apparent total tract digestibility of dry mater, gross energy and ether extract. However, other studies have reported that the AFB_1_ diet failed to affect growth performance of pigs [34,35,36]. The difference between the experimental results may be partly due to the different physiological stages (age, sex, or body weight) of pigs, the source of the contamination (purified AFB_1_ or naturally contaminated feedstuff), and the dietary concentration of AFB_1_.

Furthermore, Na+-dependent glucose transporter1 (*SGLT1*) and solute carrier family 7 member 1 (*SLC7A1*) in the small intestine epithelium are closely related to nutrient absorption. *SGLT1* is an important glucose transporter, mainly responsible for transporting luminal glucose across the intestinal epithelium [37]. *SLC7A1* is an important luminal amino acid transporter located in the intestinal mucosa [38]. In the present study, pigs fed the AFB_1_ diet showed significantly decreased mRNA levels of *SGLT1* and *SLC7A1* in the jejunal mucosa, so nutrient absorption of these pigs could be poorer than that of pigs fed the basal diet. This result is consistent with the reduction of growth performance and nutrient digestibility in the AFB_1_ group. This may be related to changes in the integrity of the intestinal barrier.

The integrity of the intestinal barrier plays a key role in the digestion and absorption of nutrients and the inhibition of pathogen invasion. Recent studies have revealed that AFB_1_ could cause remarkable disturbances in intestinal barrier function [39,40], which was supported by a study that observed the biotransformation of AFB_1_ to the toxic AFB1-exo-8,9-epoxide (AFBO) also occurred in the intestinal tract [41]. Diamine oxidase (DAO) is an intracellular enzyme located in the intestinal epithelium, which is released into the blood when the intestinal barrier is destroyed [42]. Therefore, serum DAO activity can be used as an index to evaluate intestinal permeability [43]. In the present study, pigs fed the AFB_1_ diet showed significantly increased serum DAO activity, indicating that AFB_1_ supplementation damages intestinal barrier integrity. Furthermore, the tight junction proteins (*ZO-1*, occluding and claudin-1) play an important role in regulating and maintaining intestinal permeability [44]. In the current experiment, pigs fed the AFB_1_ diet showed significantly decreased mRNA abundance of *ZO-1* in jejunal mucosa, further indicating that AFB_1_ could damage the intestinal barrier’s integrity.

Oxidative stress has been associated with intestinal barrier disruption [45]. It has been reported that AFB_1_ can initiate the production of free radicals [46], indicating the involvement of AFB_1_ in an oxidative stress pathway. SOD is a crucial antioxidant enzyme for scavenging free radicals [47]. Cao et al. reported that broilers fed an AFB_1_ contaminated diet showed significantly reduced SOD activity in the liver [48]. Similarly, our current research found that AFB_1_ supplementation significantly decreased the activity of SOD in the jejunal mucosa, indicating that AFB_1_ could decrease intestinal antioxidant ability. In addition, intestinal oxidative damage was evaluated by measuring the concentrations of PCO, 8-OHdG and MDA, indicating the degree of protein, DNA and lipid peroxidation, respectively [49]. In the present study, pigs fed the AFB_1_ diet tended to show increased content of 8-OHdG in the jejunal mucosa. The results indicated that AFB_1_ may damage intestinal barrier partially through the reduction of intestinal antioxidant ability.

Cytokines exert momentous influences on the immune and inflammatory responses and participate in the regulation of intestinal barrier integrity [50]. Previous studies have reported that pro-inflammatory cytokines (such as *TNF-α*, *IL-1β* and *IL-6*) increase intestinal permeability by inducing the disruption of tight junctions [51]. In this study, consistent with the decreased *ZO-1* mRNA levels in the jejunal mucosa of the AFB_1_ group, increased levels of the pro-inflammatory cytokines *TNF-α* and *IL-1β* mRNA abundance were observed, indicating that AFB_1_ may damage intestinal barrier integrity partially by stimulating pro-inflammatory cytokine production. However, up-regulation of anti-inflammatory cytokine *TGF-β* mRNA abundance was also observed in the AFB_1_ group. The reason may be that pigs fed the AFB_1_ diet acquired intestinal injury; in order to maintain intestinal health, the animals’ bodies increased the expression of *TGF-β* through immune mechanisms to alleviate excessive intestinal injury.

The flora in the gastrointestinal tract play an important role in the maturation of the immune system and the development of normal intestinal morphology [52]. Some degree of internal and external stimulation or interference of the body may trigger a change in the numbers or the components of intestinal microflora, cause physiochemical reactions, and lead to diseases [53]. Previous studies have reported that AFB_1_ exposure can cause gut dysbiosis and disrupt the balance of gut microbiota by increasing the growth of non-beneficial and pathogenic bacteria [54]. Oswald et al. (2003) reported that mycotoxin fumonisin B_1_ increased intestinal colonization of pathogenic *Escherichia coli* in pigs [55]. In this study, pigs fed the AFB_1_ diet showed a significantly increased *Escherichia coli* population in colonic digesta. Previous studies indicated that pathogenic *Escherichia coli* infection may damage the intestinal barrier and cause inflammatory responses in children and pigs [56]. Thus, AFB_1_-induced increase of the intestinal *Escherichia coli* population may also be an important reason that it inhibits growth and damages intestinal barrier integrity in pigs.

## 5. Conclusions

In conclusion, chronic exposure to low levels of dietary aflatoxin B_1_ suppressed growth performance, reduced the apparent total tract digestibility and damaged intestinal barrier integrity in pigs, which could be associated with the decreased intestinal antioxidant capacity and the increased pro-inflammatory cytokine production.

## Figures and Tables

**Table 1 animals-11-00336-t001:** The concentration of mycotoxin in diets.

Mycotoxins	CON ^1^	AFB_1_ ^1^	Limit ^2^
AFB_1_ (ug/kg)	0.40	286.60	20.0
ZEA (ug/kg)	ND ^3^	49.9	500
DON (ug/kg)	101.10	406.40	1000
T2 (ug/kg)	ND ^3^	ND ^3^	1000
OTA (ug/kg)	ND ^3^	ND ^3^	100

AFB1, aflatoxin B1; ZEA, zearalenone; DON, deoxynivalenol; T2, T2 toxins; OTA, ochratoxin. ^1^ CON, basal diet; AFB_1_, the basal diet supplemented with 280 μg/kg AFB_1_. ^2^ Chinese National Standard (GB) 13078-2001, GB 13078.2-2006, GB 13078.3-2007, and GB 21693-2008 of China (Beijing, China) ^3^ ND: Not detected.

**Table 2 animals-11-00336-t002:** Diet and chemical compositions of basal diets (%, as-fed basis).

Items	38 to 50 kg	50 to 75 kg	75 to 100 kg	100 to 135 kg
Ingredients				
Maize	72.11	78.68	78.64	84.41
Soybean meal, dehulled	18.14	16.76	17.42	12.04
Fish meal	3.00			
Sucrose	2.00			
Choline chloride	0.10	0.15	0.15	0.15
NaCl	0.30	0.40	0.40	0.40
Soybean oil	1.40	0.91	0.80	0.60
Limestone	0.58	0.74	0.56	0.58
CaHPO_4_	0.93	0.94	0.93	0.71
L-Lysine-HCl	0.38	0.39	0.22	0.21
DL-Methionine	0.08	0.06		
L-Threonine	0.11	0.11	0.04	0.05
L-Tryptophan	0.03	0.03		0.01
Rice	0.30	0.30	0.30	0.30
Rice bran	0.31	0.30	0.31	0.31
Vitamin premix ^1^	0.03	0.03	0.03	0.03
Mineral premix ^2^	0.20	0.20	0.20	0.20
Total	100.00	100.00	100.00	100.00
Nutrient compositions				
Metabolizable energy, MJ/kg	13.92	13.75	13.75	13.82
Crude protein	16.47	14.50	13.60	12.60
Calcium	0.66	0.59	0.52	0.46
Total phosphorus	0.58	0.50	0.50	0.44
Available phosphorus	0.32	0.25	0.25	0.21
SID ^3^ Lysine	1.10	0.96	0.84	0.70
SID ^3^ Methionine	0.37	0.30	0.25	0.22

^1^ Supplied per kilogram of diets: 12,000 IU vitamin A; 3000 IU vitamin D3; 11.23 IU vitamin E; 0.6 mg vitamin B1; 4.8 mg vitamin B2; 1.8 mg vitamin B6; 9 ug vitamin B12; 1.5 mg vitamin K3; 10.5 mg niacin; 0.15 mg folic acid; 7.5 mg pantothenic. ^2^ Supplied per kilogram of diets: 4.0 mg Cu (CuSO_4_·5H2O); 60 mg Fe (FeSO_4_·H2O); 2.0 mg Mn (MnSO_4_·H2O); 60 mg Zn (ZnSO_4_·H2O); 0.2 mg Se (Na_2_SeO_3_); 0.14 mg I (KI). ^3^ Standardized ileal digestible.

**Table 3 animals-11-00336-t003:** Primer sequences used for quantitative RT-PCR.

Gene	Sequence (5′–3′)	Product Size (bp)	Accession No.
*SGLT1*	F: GCAACAGCAAAGAGGAGCGTAT	95	NM_001164021.1
	R: GCCACAAAACAGGTCATAGGTC		
*SLC7A1*	F: CTTTCTACCCGCGGTCTCC	150	NM_001012613.1
	R: TGCTGAGCGAATCTGCTGTA		
*ZO-1*	F: CAGCCCCCGTACATGGAGA	114	XM_005659811
	R: GCGCAGACGGTGTTCATAGTT		
*Occludin*	F: CTACTCGTCCAACGGGAAAG	158	NM_001163647.2
	R: ACGCCTCCAAGTTACCACTG		
*TNF-α*	F: ACCACGCTCTTCTGCCT	121	NM_214022.1
	R: GGCTTATCTGAGGTTTG		
*IL-8*	F: AGTGGACCCCACTGTGAAAA	102	X61151.1
	R: TACAACCTTCTTCTGCACCCA		
*TGF-β*	F: AGGACCTGGGCTGGAAGTG	119	NM_214015.1
	R: GGGCCCCAGGCAGAAAT		
*IL-1β*	F: TCTGCCCTGTACCCCAACTG	112	NM_214055.1
	R: CCAGGAAGACGGGCTTTTG		
*β-actin*	F: CCACGCCCTTTCTCACTTGT	114	DQ178122
	R: CACCCACAGCACCTTATGCT		

SGLT1, sodium-glucose cotransporter 1; SLC7A1, solute carrier family 7 member 1; ZO-1, zonula occluden-1; TNF-α, tumor necrosis factor-α; IL-8, interleukin-8; TGF-β, transforming growth factor-β; IL-1β, interleukin-1β.

**Table 4 animals-11-00336-t004:** Primer and probe sequences used for quantitative RT-PCR.

Items	Sequence (5′–3′)	Anneal Temperature (°C)	Product Size (bp)
Total bacteria	F: ACTCCTACGGGAGGCAGCAG R: ATTACCGCGGCTGCTGG	60.0	200
*Lactobacillus*	F: GAGGCAGCAGTAGGGAATCTTC R: CAACAGTTACTCTGACACCCGTTCTTC P: AAGAAGGGTTTCGGCTCGTAAAACTCTGTT	57.5	126
*Bifidobacterium*	F: CGCGTCCGGTGTGAAAG R: CTTCCCGATATCTACACATTCCA P: ATTCCACCGTTACACCGGAA	59.5	121
*Bacillus*	F: GCAACGAGCGCAACCCTTGA R: TCATCCCCACCTTCCTCCGGT P: CGGTTTGTCACCGGCAGTCACCT	60.0	92
*Escherichia coli*	F: CATGCCGCGTGTATGAAGAA R: CGGGTAACGTCAATGAGCAAA P: AGGTATTAACTTTACTCCCTTCCTC	58.8	96

**Table 5 animals-11-00336-t005:** Effects of AFB_1_ on growth performance of pigs.

Items	CON	AFB_1_	*p*-Value
Initial BW (kg)	38.22 ± 0.70	38.19 ± 0.65	0.98
Final BW (kg)	132.80 ± 2.10	124.60 ± 3.43	0.07
ADFI (g/d)	2544.08 ± 41.64	2332.18 ± 96.26	0.07
ADG (g/d)	927.25 ± 15.69	847.16 ± 33.52	0.06
F/G	2.75 ± 0.04	2.75 ± 0.05	0.91

Results are expressed as means ± standard errors (*n* = 7). BW, body weight; ADFI, average daily feed intake; ADG, average daily gain; F/G, the ratio of feed intake to gain. CON, basal diet; AFB_1_, the basal diet supplemented with 280 μg/kg AFB_1_.

**Table 6 animals-11-00336-t006:** Effects of AFB_1_ on the apparent total tract digestibility of pigs (%).

Items	CON	AFB_1_	*p*-Value
50 to 75 kg			
Dry mater	87.74 ± 0.77	86.17 ± 0.44	0.04
Gross energy	87.34 ± 0.76	85.66 ± 0.45	0.03
Crude protein	85.22 ± 1.18	84.62 ± 0.77	0.61
Ether extract	75.99 ± 2.19	73.63 ± 1.22	0.25
75 to 105 kg			
Dry mater	89.59 ± 0.19	89.82 ± 0.58	0.76
Gross energy	89.08 ± 0.22	89.1 ± 0.57	0.98
Crude protein	87.24 ± 0.44	87.63 ± 1.1	0.80
Ether extract	81.88 ± 0.43	78.06 ± 0.74	0.04
105 to 135 kg			
Dry mater	88.25 ± 0.36	86.39 ± 0.60	0.03
Gross energy	87.76 ± 0.34	85.54 ± 0.58	0.01
Crude protein	84.80 ± 1.18	83.12 ± 0.79	0.27
Ether extract	73.88 ± 1.78	71.99 ± 1.47	0.43

Results are expressed as means ± standard errors (*n* = 7). CON, basal diet; AFB_1_, the basal diet supplemented with 280 μg/kg AFB_1_.

**Table 7 animals-11-00336-t007:** Effects of AFB_1_ on the relative mRNA expressions of nutrient transporters in jejunal mucosa of pigs.

Items	CON	AFB_1_	*p*-Value
*SGLT1*	1.00 ± 0.19	0.36 ± 0.12	0.02
*SLC7A1*	1.00 ± 0.03	0.86 ± 0.04	0.04

Results are expressed as means ± standard errors (*n* = 7). SGLT1, sodium-glucose cotransporter 1; SLC7A1, solute carrier family 7 member 1. CON, basal diet; AFB_1_, the basal diet supplemented with 280 μg/kg AFB_1_.

**Table 8 animals-11-00336-t008:** Effects of AFB_1_ on serum DAO activity and the relative mRNA expressions of barrier junction-related genes in jejunal mucosa.

Items	CON	AFB_1_	*p*-Value
Serum			
DAO (U/L)	13.79 ± 1.97	23.75 ± 1.65	*p* < 0.01
Jejunal mucosa			
*ZO-1*	1.00 ± 0.05	0.67 ± 0.03	*p* < 0.01
*Occludin*	1.00 ± 0.12	1.06 ± 0.08	0.68

Results are expressed as means ± standard errors (*n* = 7). DAO, diamine oxidase; ZO-1, zonula occluden-1. CON, basal diet; AFB_1_, the basal diet supplemented with 280 μg/kg AFB_1_.

**Table 9 animals-11-00336-t009:** Effects of AFB_1_ on jejunal mucosal antioxidant indicators of pigs.

Items	CON	AFB_1_	*p*-Value
T-AOC (U/mgprot)	1.44 ± 0.10	1.20 ± 0.16	0.24
SOD (U/mgprot)	275.34 ± 21.06	189.34 ± 18.62	0.02
MDA (nmol/mgprot)	0.18 ± 0.02	0.18 ± 0.03	0.95
8-OHdG (pg/mL)	11.22 ± 0.96	15.03 ± 1.63	0.08
PCO (pg/mL)	26.80 ± 2.28	31.47 ± 3.67	0.31

Results are expressed as means ± standard errors (*n* = 7). T-AOC, total antioxidant capacity; SOD, superoxide dismutase; MDA, malondialdehyde; 8-OHdG, 8-hydroxy-2′-deoxyguanosine; PCO, protein carbonylation. CON, basal diet; AFB_1_, the basal diet supplemented with 280 μg/kg AFB_1_.

**Table 10 animals-11-00336-t010:** Effects of AFB1 on the relative mRNA expressions of inflammatory related genes in jejunal mucosa of pigs.

Items	CON	AFB_1_	*p*-Value
*TNF-α*	1.00 ± 0.08	1.44 ± 0.14	0.03
*IL-1β*	1.00 ± 0.06	1.34 ± 0.10	0.02
*IL-8*	1.00 ± 0.11	1.42 ± 0.28	0.21
*TGF-β*	1.00 ± 0.08	1.56 ± 0.21	0.05

Results are expressed as means ± standard errors (*n* = 7). TNF-α, tumor necrosis factor-α; IL-1β, interleukin-1β; IL-8, interleukin-8; TGF-β, transforming growth factor-β. CON, basal diet; AFB_1_, the basal diet supplemented with 280 μg/kg AFB_1_.

**Table 11 animals-11-00336-t011:** Effects of AFB_1_ on bacteria populations in colonic digesta of pigs (log10(copies/g)).

Items	CON	AFB_1_	*p*-Value
Total bacteria	13.46 ± 0.09	13.54 ± 0.02	0.45
*Lactobacillus*	7.74 ± 0.24	7.82 ± 0.23	0.82
*Bacillus*	9.84 ± 0.17	9.93 ± 0.05	0.63
*Escherichia coli*	6.72 ± 0.37	7.72 ± 0.13	0.03
*Bifidobacterium*	5.43 ± 0.13	5.62 ± 0.35	0.63

Results are expressed as means ± standard errors (*n* = 7). CON, basal diet; AFB_1_, the basal diet supplemented with 280 μg/kg AFB_1_.

## Data Availability

No new data were created or analyzed in this study. Data sharing is not applicable to this article.

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
