# Peer review of "Effects of Chronic Exposure to Low Levels of Dietary Aflatoxin B1 on Growth Performance, Apparent Total Tract Digestibility and Intestinal Health in Pigs"

_animals, 2021, doi:10.3390/ani11020336_

Round 1

Reviewer 1 Report

This manuscript has investigated the effect of effects of chronic exposure of low-level of aflatoxin b1 on growth performance, nutrient digestibility and intestinal health in pigs. Discussion is well-written but there are several places that need revision. Please check the usage of ‘the’.

L2 revise to ‘chronic exposure to low level of dietary AFB1 on…’ or ‘exposure of a diet with…’

L11 revise to ‘one of the most toxic mycotoxin compounds’

L12 Aspergillus -> italic throughout the manuscript (bacteria names), revise to ‘a common fungi contaminant…’

L13-14 please revise this sentence.

L15 revise to ‘…is common in swine production and…’

L16 revise to ‘chronic exposure to low level of dietary AFB1…’

L18, 22 , 37 AFB1 instead of aflatoxin B1

L23 revise to ‘exposure to low level of dietary aflatoxin B1 (AFB1)……apparent total tract digestibility…’

L25 please move breed information into the parenthesis with initial body weight.

L25 revise to ‘into control (CON, basal diet) and AFB1 groups…’

L27 revise to ‘…the AFB1 exposure..’

L28 please use ‘apparent total tract digestibility’ instead of just ‘nutrient digestibility’ or ‘digestibility’ throughout the manuscript except in Simple Summary’.

If a word was used only one time, don’t use acronym in the abstract (DAO, SGLT1…).

L36 Escherichia coli -> italic

L36-40 revise to ‘chronic exposure to low level of dietary AFB1..’ throughout the manuscript. Also revise to ‘…suppressed growth…reduced…and damaged…which could be associated…’

Do not use an acronym at the beginning of the sentence.

L50 revise to ‘…to be hepatotoxic…-genic…immunosuppressive ’

L63 please add reference range of aflatoxin content in swine feed or maximum tolerance level.

Table 1: please report ME instead of DE. Soybean meal dehulled or hulled or CP level? Calculated composition for lysine and methionine is what basis? SID? Please add vitamins and trace mineral sources.

Please provide average body weight of each phase when fecal samples were collected.

L103 revise to ‘recording feed disappearance and waste, and weighing pigs.’

L104 revise to ‘Diet and chemical compositions of basal diets…’

L110 revise to ‘Apparent total tract digestibility…’, please add initial body weight and day of experiment when the fecal collection occurred.

L111 fecal samples were collected daily? From all pigs in each group?  Please add information how feces were thawed, dried, and ground.

L113 revise to ‘…until chemical analysis for crude protein…

L117 form -> from

L117 pigs were fasted before slaughter?

L121-121 please add digestibility calculation equation. Please add AIA analysis method.

L129 remove ‘levels’

No data of beta actin.

L166-170 Please add model terms, one-way ANOVA and used Student’s t-test for mean separation.

L172 Were there data of body weight, ADG, ADFI and F:G in each phase? Since digestibility was obtained in each phase (except 38-50 kg) and had different response, it would be better to have growth performance in each phase.

Please add treatment description in each table.

Table 5 and L178-182 apparent total tract digestibility. Please move DM digestibility to the top, then gross energy, crude protein and ether extract in order.

Table 6 revise to ‘…relative mRNA expressions of nutrient transporters in jejunal mucosa of pigs’.

L185 revise to ‘Relative mRNA expressions of nutrient transporters in jejunal mucosa’

L186 revise to ‘…the AFB1 diet significantly decreased mRNA expression of …compared with those fed the CON diet.’

L190 please revise the title of 3.4 and Table 7 by reflecting the comment for Table 6.

Each table for nutrient transporter, DAO, barrier function, antioxidant capacity, and cytokines should need more information on the footnote such as time of sample collection, each acronym, relative values or not, etc. mRNA expression data are all relative comparisons.

L211-214 Italic on bacteria name.

L219 revise to ‘health issues including…’

L223 revise to ‘aflatoxin decreased growth rate and feed intake…’

L224 revise to ‘a decrease in growth rate of piglets fed…’

L238 revise to ‘…pigs fed the AFB1 diet…so nutrient absorption of these pigs could be poorer…fed the basal diet.’

L239 revise to ‘growth performance and nutrient digestibility…’

L281 Oswald et al. (2003)

L289-292 suppressed, reduced, damaged…

Reviewer 2 Report

The study aimed to investigate the chronic effect of dietary Aflatoxins on growth performance, nutrient digestibility and intestinal health in growing pigs.

Some revisions are needed in M&M section, and how authors showed the results.

English language should be revised by professional reviewer, there are grammatical errors.

Line 23: please revise English 

Line 29: lipids?

Line 30-31: mRNA of which tissue?

Line 62: "intestinal health" of pigs?

Line 88: where? in the diet? did authors perform analysis in both diet control and treated?

Line 94-95: pleas define these limits 

Line 94-95: pleas define these limits 

Line 102 please deled 2012

Line 103: please specify when authors recorded ADFI and ADG, day, week month? 

I suggest to use experimental days instead pigs weight ( 38 to 50, 50 to 75), make very difficult to understand and follow the experiment. 

Line 110: please define " each stage" 

Line 114: please modulate the phrase, better " Blood samples were collected from.." 

Line 126: please briefly explain the method 

Line 126: please briefly explain the method 

Line 150: I am concern to use only one Housekeeping genes, why authors did not use two of three HKG? this procedure should be better with two and three. 

Table 5: Please better separate the different stages in the table, adding a row or using bold fold 

Line 223: " aflatoxinsa"

Conclusions: I suggest to improve the conclusion of the study

Round 2

Reviewer 1 Report

Thanks for the revision.

L84 Italic (flavus).